# (Probably) Concave Graph Matching

**Haggai Maron**
Weizmann Institute of Science
Rehovot, Israel
haggai.maron@weizmann.ac.il

**Yaron Lipman**
Weizmann Institute of Science
Rehovot, Israel
yaron.lipman@weizmann.ac.il

## Abstract

In this paper, we address the graph matching problem. Following the recent works of Zaslavskiy et al. (2009); Vestner et al. (2017) we analyze and generalize the idea of concave relaxations. We introduce the concepts of *conditionally concave* and *probably conditionally concave* energies on polytopes and show that they encapsulate many instances of the graph matching problem, including matching Euclidean graphs and graphs on surfaces. We further prove that local minima of probably conditionally concave energies on general matching polytopes (*e.g.*, doubly stochastic) are with high probability extreme points of the matching polytope (*e.g.*, permutations).

## 1 Introduction

Graph matching is a generic and popular modeling tool for problems in computational sciences such as computer vision (Berg et al., 2005; Zhou and De la Torre, 2012; Rodola et al., 2013; Bernard et al., 2017), computer graphics (Funkhouser and Shilane, 2006; Kezurer et al., 2015), medical imaging (Guo et al., 2013), and machine learning (Umeyama, 1988; Huet et al., 1999; Cour et al., 2007). In general, graph matching refers to several different optimization problems of the form:

$$\min_{X} \ E(X) \quad \text{s.t.} \quad X \in \mathcal{F} \tag{1}$$

where $\mathcal{F} \subset \mathbb{R}^{n \times n_0}$ is a collection of *matchings* between vertices of two graphs $G_A$ and $G_B$, and $E(X) = [X]^T M[X] + a^T[X]$ is usually a quadratic function in $X \in \mathbb{R}^{n \times n_0}$ ($[X] \in \mathbb{R}^{nn_0 \times 1}$ is its column stack). Often, $M$ quantifies the discrepancy between edge affinities exerted by the matching $X$. Edge affinities are represented by symmetric matrices $A \in \mathbb{R}^{n \times n}$, $B \in \mathbb{R}^{n_0 \times n_0}$. Maybe the most common instantiation of (1) is

$$E_1(X) = \|AX - XB\|_F^2 \tag{2}$$

and $\mathcal{F} = \Pi_n$, the matrix group of $n \times n$ permutations. The permutations $X \in \Pi_n$ represent bijections between the set of $(n)$ vertices of $G_A$ and the set of $(n)$ vertices of $G_B$. We denote this problem as GM. From a computational point of view, this problem is equivalent to the quadratic assignment problem, and as such is an NP-hard problem (Burkard et al., 1998). A popular way of obtaining approximate solutions is by relaxing its combinatorial constraints (Loiola et al., 2007).

A standard relaxation of this formulation (*e.g.* Almohamad and Duffuaa (1993); Aflalo et al. (2015); Fiori and Sapiro (2015)) is achieved by replacing $\Pi_n$ with its convex hull, namely the set of doubly-stochastic matrices $\mathrm{DS} = \mathrm{hull}(\mathcal{F}) = \left\{ X \in \mathbb{R}^{n \times n} \mid X\mathbf{1} = \mathbf{1}, X^T\mathbf{1} = \mathbf{1}, X \geq 0 \right\}$. The main advantage of this formulation is the convexity of the energy $E_1$; the main drawback is that often the minimizer is not a permutation and simply projecting the solution onto $\Pi_n$ doesn't take the energy into account resulting in a suboptimal solution. The prominent Path Following algorithm (Zaslavskiy et al., 2009) suggests a better solution of continuously changing $E_1$ to a concave energy $E'$ that coincide (up to an additive constant) with $E_1$ over the permutations. The concave energy $E'$ is called *concave relaxation* and enjoys three key properties: (i) Its solution set is the same as the GM problem. (ii) Its set of local optima are all permutations. This means no projection of the local optima onto the permutations is required. (iii) For every descent direction, a maximal step is always guaranteed to reduce the energy most.

Dym et al. (2017); Bernard et al. (2017) suggest a similar strategy but starting with a tighter convex relaxation. Another set of works (Vogelstein et al., 2015; Lyzinski et al., 2016; Vestner et al., 2017; Boyarski et al., 2017) have considered the energy

$$E_2(X) = -\text{tr}(BX^T AX) \tag{3}$$

over the doubly-stochastic matrices, DS, as well. Note that both energies $E_1$, $E_2$ are identical (up to an additive constant) over the permutations and hence both are considered relaxations. However, in contrast to $E_1$, $E_2$ is in general indefinite, resulting in a non-convex relaxation. Vogelstein et al. (2015); Lyzinski et al. (2016) suggest to locally optimize this relaxation with the Frank-Wolfe algorithm and motivate it by proving that for the class of $\rho$-correlated Bernoulli adjacency matrices $A, B$, the optimal solution of the relaxation almost always coincides with the (unique in this case) GM optimal solution. Vestner et al. (2017); Boyarski et al. (2017) were the first to make the useful observation that $E_2$ is *itself a concave relaxation* for some important cases of affinities such as heat kernels and Gaussians. This leads to an efficient local optimization using the Frank-Wolfe algorithm and specialized linear assignment solvers (*e.g.*, Bernard et al. (2016)).

In this paper, we analyze and generalize the above works and introduce the concepts of *conditionally concave* and *probably conditionally concave* energies $E(X)$. Conditionally concave energy $E(X)$ means that the restriction of the Hessian $M$ of the energy $E$ to the linear space

$$\text{lin(DS)} = \left\{ X \in \mathbb{R}^{n \times n} \mid X\mathbf{1} = 0, X^T\mathbf{1} = 0 \right\} \tag{4}$$

is negative definite. Note that $\text{lin(DS)}$ is the linear part of the affine-hull of the doubly-stochastic matrices, denoted $\text{aff(DS)}$. We will use the notation $M|_{\text{lin(DS)}}$ to refer to this restriction of $M$, and consequently $M|_{\text{lin(DS)}} \prec 0$ means $v^T M v < 0$, for all $0 \neq v \in \text{lin(DS)}$. Our first result is proving there is a large class of affinity matrices resulting in conditionally concave $E_2$. In particular, affinity matrices constructed using *positive or negative definite functions*[1] will be conditionally concave.

**Theorem 1.** *Let $\Phi : \mathbb{R}^d \to \mathbb{R}$, $\Psi : \mathbb{R}^s \to \mathbb{R}$ be both conditionally positive (or negative) definite functions of order 1. For any pair of graphs with affinity matrices $A, B \in \mathbb{R}^{n \times n}$ so that*

$$A_{ij} = \Phi(x_i - x_j), \quad B_{ij} = \Psi(y_i - y_j) \tag{5}$$

*for some arbitrary $\{x_i\}_{i \in [n]} \subset \mathbb{R}^d$, $\{y_i\}_{i \in [n]} \subset \mathbb{R}^s$, the energy $E_2(X)$ is conditionally concave, i.e., its Hessian $M|_{\text{lin(DS)}} \prec 0$.*

One useful application of this theorem is in matching graphs with Euclidean affinities, since Euclidean distances are conditionally negative definite of order 1 (Wendland, 2004). That is, the affinities are Euclidean distances of points in Euclidean spaces of arbitrary dimensions,

$$A_{ij} = \|x_i - x_j\|_2, \quad B_{ij} = \|y_i - y_j\|_2, \tag{6}$$

where $\{x_i\}_{i \in [n]} \subset \mathbb{R}^d$, $\{y_i\}_{i \in [n]} \subset \mathbb{R}^s$. This class contains, besides Euclidean graphs, also affinities made out of distances that can be isometrically embedded in Euclidean spaces such as diffusion distances (Coifman and Lafon, 2006), distances induced by deep learning embeddings (*e.g.* Schroff et al. (2015)) and Mahalanobis distances. Furthermore, as shown in Bogomolny et al. (2007) the spherical distance, $A_{ij} = d_{S^d}(x_i, x_j)$, is also conditionally negative definite over the sphere and therefore can be used in the context of the theorem as-well.

Second, we generalize the notion of conditionally concave energies to *probably conditionally concave* energies. Intuitively, the energy $E$ is called *probably conditionally concave* if it is rare to find a linear subspace $D$ of $\text{lin(DS)}$ so that the restriction of $E$ to it is convex, that is $M|_D \succeq 0$. The primary motivation in considering probably conditionally concave energies is that they enjoy (with high probability) the same properties as the conditionally concave energies, *i.e.*, (i)-(iii). Therefore, locally minimizing probably conditionally concave energies over $\mathcal{F}$ can be done also with the Frank-Wolfe algorithm, with guarantees (in probability) on the feasibility of both the optimization result and the solution set of this energy.

A surprising fact we show is that probably conditionally concave energies are pretty common and include Hessian matrices $M$ with almost the same ratio of positive to negative eigenvalues. The

following theorem bounds the probability of finding uniformly at random a linear subspace $D$ such that the restriction of $M \in \mathbb{R}^{m \times m}$ to $D$ is convex, *i.e.*, $M|_D \succ 0$. The set of $d$-dimensional linear subspaces of $\mathbb{R}^m$ is called the Grassmannian $G_r(d, m)$ and it has a compact differential manifold structure and a uniform measure $P_r$.

**Theorem 2.** *Let $M \in \mathbb{R}^{m \times m}$ be a symmetric matrix with eigenvalues $\lambda_1, \ldots, \lambda_m$. Then, for all $t \in (0, \frac{1}{2\lambda_{\max}})$:*

$$P_r(M|_D \succeq 0) \leq \prod_{i=1}^{m}(1 - 2t\lambda_i)^{-\frac{d}{2}}, \tag{7}$$

*where $M|_D$ is the restriction of $M$ to the d-dimensional linear subspace defined by $D \in G_r(d, m)$ and the probability is taken with respect to the Haar probability measure on $G_r(d, m)$.*

For the case $d = 1$ the probability of $M|_D \succeq 0$ can be interpreted via distributions of quadratic forms. Previous works aimed at calculating and bounding similar probabilities (Imhof, 1961; Rudelson et al., 2013) but in different (more general) settings providing less explicit bounds. As we will see, the case $d > 1$ quantifies the chances of local minima residing at high dimensional faces of $\text{hull}(\mathcal{F})$.

As a simple use-case of theorem 2, consider a matrix where $51\%$ of the eigenvalues are $-1$ and $49\%$ are $+1$; the probability of finding a convex direction of this matrix, when the direction is uniformly distributed, is exponentially low in the dimension of the matrix. As we (empirically) show, one class of problems that in practice presents probably conditionally concave $E_2$ are when the affinities $A, B$ describe geodesic distances on surfaces.

Probable concavity can be further used to prove theorems regarding the likelihood of finding a local minimum outside the matching set $\mathcal{F}$ when minimizing $E$ over a relaxed matching polytope $\text{hull}(\mathcal{F})$. We will show the existence of a rather general probability space (in fact, a family) $(\Omega_m, P_r)$ of Hessians $M \in \mathbb{R}^{m \times m} \in \Omega_m$ with a natural probability measure, $P_r$, so that the probability of local minima of $E(X)$ to be outside $\mathcal{F}$ is very small. This result is stated and proved in theorem 3. An immediate conclusion of this result provides a proof of a probabilistic version of properties (i) and (ii) stated above for energies drawn from this distribution. In particular, the global minima of $E(X)$ over DS coincide with those over $\Pi_n$ with high probability. The following theorem provides a general result in the flavor of Lyzinski et al. (2016) for a large class of quadratic energies.

**Theorem 4.** *Let $E$ be a quadratic energy with Hessian drawn from the probability space $(\Omega_m, P_r)$. The chance that a local minimum of $\min_{X \in \text{DS}} E(X)$ is outside $\Pi_n$ is extremely small, bounded by $exp(-c_1 n^2)$, for some constant $c_1 > 0$.*

Third, when the energy of interest $E(X)$ is not probably conditionally concave over $\text{lin}(\mathcal{F})$ there is no guarantee that the local optimum of $E$ over $\text{hull}(\mathcal{F})$ is in $\mathcal{F}$. We devise a simple variant of the Frank-Wolfe algorithm, replacing the standard line search with a *concave search*. Concave search means subtracting from the energy $E$ convex parts that are constant on $\mathcal{F}$ (*i.e.*, relaxations) until an energy reducing step is found.

## 2 Conditionally concave energies

We are interested in the application of the Frank-Wolfe algorithm Frank and Wolfe (1956) for locally optimizing $E_2$ (potentially with a linear term) from (3) over the doubly-stochastic matrices:

$$\min_{X} \quad E(X) \tag{8a}$$

$$\text{s.t.} \quad X \in \text{DS} \tag{8b}$$

where $E(X) = -[X]^T(B \otimes A)[X] + a^T[X]$. For completeness, we include a simple pseudo-code:

---
**input:** $X_0 \in \text{hull}(\mathcal{F})$

**while** *not converged* **do**
  *compute step:* $X_1 = \min_{X \in \text{DS}} -2[X_0]^T(B \otimes A)[X] + a^T[X]$;
  *line-search:* $t_0 = \text{argmin}_{t \in [0,1]} E((1-t)X_0 + tX_1)$ ;
  *apply step:* $X_0 = (1-t_0)X_0 + t_0 X_1$ ;
**end**

---
**Algorithm 1:** Frank-Wolfe algorithm.

**Definition 1.** *We say that $E(X)$ is conditionally concave if it is concave when restricted to the linear space $\text{lin}(\mathcal{F})$, the linear part of the affine-hull $\text{hull}(\mathcal{F})$.*

If $E(X)$ is conditionally concave we have that properties (i)-(iii) of concave relaxations detailed above hold. In particular Algorithm 1 would always accept $t_0 = 1$ as the optimal step, and therefore it will produce a series of feasible matchings $X_0 \in \Pi_n$ and will converge after a finite number of steps to a permutation local minimum $X_* \in \Pi_n$ of (8). Our first result in this paper provides sufficient condition for $W = -B \otimes A$ to be concave. It provides a connection between *conditionally positive (or negative) definite* functions (Wendland, 2004), and negative definiteness of $-B \otimes A$:

**Definition 2.** *A function $\Phi : \mathbb{R}^d \to \mathbb{R}$ is called* conditionally positive definite of order $m$ *if for all pairwise distinct points $\{x_i\}_{i \in [n]} \subset \mathbb{R}^d$ and all $0 \neq \eta \in \mathbb{R}^n$ satisfying $\sum_{i \in [n]} \eta_i p(x_i) = 0$ for all $d$-variate polynomials $p$ of degree less than $m$, we have $\sum_{ij=1}^n \eta_i \bar\eta_j \Phi(x_i - x_j) > 0$.*

Specifically, $\Phi$ is conditionally positive definite of order 1 if for all pairwise distinct points $\{x_i\}_{i \in [n]} \subset \mathbb{R}^d$ and zero-sum vectors $0 \neq \eta \in \mathbb{R}^d$ we have $\sum_{ij=1}^n \eta_i \bar\eta_j \Phi(x_i - x_j) > 0$. Conditionally negative definiteness is defined analogously. Some well-known functions satisfy the above conditions, for example: $-\|x\|_2$, $-(c^2 + \|x\|_2^2)^\beta$ for $\beta \in (0,1]$ are conditionally positive definite of order 1, while the functions $\exp(-\tau^2 \|x\|_2^2)$ for all $\tau$, and $c_{30} = (1 - \|x\|_2^2)_+$ are conditionally positive definite of order 0 (also called just positive definite functions). Note that if $\Phi$ is conditionally positive definite of order $m$, it is also conditionally positive definite of any order $m' > m$. Lastly, as shown in Bogomolny et al. (2007), spherical distances $-d(x, x')^\gamma$ are conditionally positive semidefinite for $\gamma \in (0,1]$, and $\exp(-\tau^2 d(x, x')^\gamma)$ are positive definite for $\gamma \in (0,1]$ and all $\tau$. We now prove:

**Theorem 1.** *Let $\Phi : \mathbb{R}^d \to \mathbb{R}$, $\Psi : \mathbb{R}^s \to \mathbb{R}$ be both conditionally positive (or negative) definite functions of order 1. For any pair of graphs with affinity matrices $A, B \in \mathbb{R}^{n \times n}$ so that*

$$A_{ij} = \Phi(x_i - x_j), \quad B_{ij} = \Psi(y_i - y_j) \tag{9}$$

*for some arbitrary $\{x_i\}_{i \in [n]} \subset \mathbb{R}^d$, $\{y_i\}_{i \in [n]} \subset \mathbb{R}^s$, the energy $E_2(X)$ is conditionally concave, i.e., its Hessian $M|_{\mathrm{lin(DS)}} \prec 0$.*

**Lemma 1** (orthonormal basis for lin(DS))**.** *If the columns of $F \in \mathbb{R}^{n \times (n-1)}$ constitute an orthonormal basis for the linear space $\mathbf{1}^\perp = \{x \in \mathbb{R}^n \mid x^T \mathbf{1} = 0\}$ then the columns of $F \otimes F$ are an orthonormal basis for $\mathrm{lin(DS)}$.*

*Proof.* First, $(F \otimes F)^T (F \otimes F) = (F^T \otimes F^T)(F \otimes F) = (F^T F) \otimes (F^T F) = I_{n-1} \otimes I_{n-1} = I_{(n-1)^2}$. Therefore $F \otimes F$ is full rank with $(n-1)^2$ orthonormal columns. Any column of $F \otimes F$ is of the form $F_i \otimes F_j$, where $F_i, F_j$ are the $i^{\text{th}}$ and $j^{\text{th}}$ columns of $F$, respectively. Now, reshaping $F_i \otimes F_j$ back into an $n \times n$ matrix using the inverse of the bracket operation we get $X = ]F_i \otimes F_j[= F_j F_i^T$ which are clearly in $\mathrm{lin(DS)}$. Lastly, since the dimension of $\mathrm{lin(DS)}$ is $(n-1)^2$ the lemma is proved. $\square$

*Proof.* (of Theorem 1 ) Let $A, B \in \mathbb{R}^{n \times n}$ be as in the theorem statement. Checking that $E(X)$ is conditionally concave amounts to restricting the quadratic form $-[X]^T (B \otimes A)[X]$ to $\mathrm{lin(DS)}$: $-(F \otimes F)^T (B \otimes A)(F \otimes F) = -(F^T B F) \otimes (F^T A F) \prec 0$, where we used Lemma 1 and the fact that $\Phi, \Psi$ are conditionally positive definite of order 1. $\square$

**Corollary 1.** *Let $A, B$ be Euclidean distance matrices then the solution set of Problem (8) and GM coincide.*

# 3 Probably conditionally concave energies

Although Theorem 1 covers a rather wide spectrum of instantiations of Problem (8) it definitely does not cover all interesting scenarios. In this section we would like to consider a more general energy $E(X) = [X]^T M [X] + a^T [X]$, $X \in \mathbb{R}^{n \times n}$, $M \in \mathbb{R}^{n^2 \times n^2}$ and the optimization problem:

$$\min_X \quad E(X) \tag{10a}$$

$$\text{s.t.} \quad X \in \mathrm{hull}(\mathcal{F}) \tag{10b}$$

We assume that $\mathcal{F} = \mathrm{ext}(\mathrm{hull}(\mathcal{F}))$, namely, the matchings are extreme points of their convex hull (as happens *e.g.*, for permutations $F = \Pi_n$). When the restricted Hessians $M|_{\mathrm{lin}(\mathcal{F})}$ are $\epsilon-$*negative definite* (to be defined soon) we will call $E(X)$ *probably conditionally concave*.

Probably conditionally concave energies $E(X)$ will possess properties (i)-(iii) of conditionally concave energies with high probability. Hence they allow using Frank-Wolfe algorithms, such as Algorithm 1, with no line search ($t_0 = 1$) and achieve local minima in $\mathcal{F}$ (no post-processing is required). In addition, we prove that certain classes of probably conditionally concave relaxations have no local minima that are outside $\mathcal{F}$, with high probability. In the experiment section we will also demonstrate that in practice this algorithm works well for different choices of probably conditionally concave energies. Popular energies that fall into this category are, for example, (3) with $A, B$ geodesic distance matrices or certain functions thereof.

We first make some preparations. Recall the definition of the *Grassmannian* $G_r(d, m)$: It is the set of $d$-dimensional linear subspaces in $\mathbb{R}^m$; it is a compact differential manifold defined by the quotient $O(m)/O(d) \times O(m - d)$, where $O(s)$ is the orthogonal group in $\mathbb{R}^s$. The orthogonal group $O(m)$ acts transitively on $G_r(d, m)$ by taking an orthogonal basis of any $d$-dimensional linear subspace to an orthogonal basis of a possibly different $d$-dimensional subspace. On $O(m)$ there exists Haar probability measure, that is a probability measure invariant to actions of $O(m)$. The Haar probability measure on $O(m)$ induces an $O(m)$-invariant (which we will also call Haar) probability measure on $G(k, m)$. We now introduce the notion of $\epsilon$-negative definite matrices:

**Definition 3.** *A symmetric matrix $M \in \mathbb{R}^{m \times m}$ is called $\epsilon$-negative* definite *if the probability of finding a d-dimensional linear subspace $D \in G(d, m)$ so that $A$ is convex over $D$ is smaller than $\epsilon^d$. That is, $P_r(\{M|_D \succeq 0\}) \leq \epsilon^d$ where the probability is taken with respect to a Haar $O(m)$-invariant measure on the Grassmannian $G_r(d, m)$.*

One way to interpret $M|_D$, the restriction of the matrix $M$ to the linear subspace $D$, is to consider a matrix $F \in \mathbb{R}^{m \times d}$ where the columns of $F$ form a basis to $D$ and consider $M|_D = F^T M F$. Clearly, negative definite matrices are $\epsilon$-negative definite for all $\epsilon > 0$. The following theorem helps to see what else this definition encapsulates:

**Theorem 2.** *Let $M \in \mathbb{R}^{m \times m}$ be a symmetric matrix with eigenvalues $\lambda_1, \ldots, \lambda_m$. Then, for all $t \in (0, \frac{1}{2\lambda_{\max}})$:*

$$P_r(M|_D \succeq 0) \leq \prod_{i=1}^{m}(1 - 2t\lambda_i)^{-\frac{d}{2}}, \tag{11}$$

*where $M|_D$ is the restriction of $M$ to the d-dimensional linear subspace defined by $D \in G_r(d, m)$ and the probability is taken with respect to the Haar probability measure on $G_r(d, m)$.*

*Proof.* Let $F$ be an $m \times d$ matrix of i.i.d. standard normal random variables $\mathcal{N}(0, 1)$. Let $F_j$, $j \in [d]$, denote the $j^{\text{th}}$ column of $F$. The multivariate distribution of $F$ is $O(m)$-invariant in the sense that for a subset $\mathcal{A} \subset \mathbb{R}^{m \times d}$, $P_r(R\mathcal{A}) = P_r(\mathcal{A})$ for all $R \in O(m)$. Therefore, $P_r(M|_D \succeq 0) = P_r(F^T M F \succeq 0)$. Next, $P_r(F^T M F \succeq 0) \leq P_r(\cap_{j=1}^d \{F_j^T M F_j \geq 0\}) = \prod_{j=1}^d P_r(F_j^T M F_j \geq 0)$, where the inequality is due to the fact that a positive semidefinite matrix necessarily has non-negative diagonal, and the equality is due to the independence of the random variables $F_j^T M F_j$, $j \in [d]$. We now calculate the probability $P_r(F_1^T M F_1)$ which is the same for all columns $j \in [d]$. For brevity let $X = (X_1, X_2, \ldots, X_m)^T = F_1$. Let $M = U\Lambda U^T$, where $U \in O(m)$ and $\Lambda = \text{diag}(\lambda_1, \lambda_2, \ldots, \lambda_m)$ be the spectral decomposition of $M$. Since $UX$ has the same distribution as $X$ we have that $P_r(X^T M X \geq 0) = P_r(X^T \Lambda X \geq 0) = P_r(\sum_{i=1}^m \lambda_i X_i^2 \geq 0)$. Since $X_i^2 \sim \chi^2(1)$ we have transformed the problem into a non-negativity test of a linear combination of chi-squared random variables. Using the Chernoff bound we have for all $t > 0$:

$$P_r\left(\sum_{i=1}^m \lambda_i X_i^2 \geq 0\right) \leq \mathbb{E}\left(e^{t \sum_{i=1}^m \lambda_i X_i^2}\right) = \prod_{i=1}^m \mathbb{E}\left[e^{t\lambda_i X_i^2},\right]$$

where the last equality follows from the independence of $X_1, ..., X_m$. To finish the proof we note that $\mathbb{E}\left[e^{t\lambda_i X_i^2}\right]$ is the moment generating function of the random variable $X_i^2$ sampled at $t\lambda_i$ which is known to be $(1 - 2t\lambda_i)^{-1/2}$ for $t\lambda_i < \frac{1}{2}$ which means that we can take $t < \frac{1}{2\lambda_i}$ when $\lambda_i \neq 0$ and disregard all $\lambda_i = 0$. $\square$

Theorem 2 shows that there is a *concentration of measure* phenomenon when the dimension $m$ of the matrix $M$ increases. For example consider

$$\Lambda_{m,p} = (\overbrace{\lambda_1, \lambda_2, \ldots,}^{(1-p)m} \overbrace{\mu_1, \mu_2, \ldots}^{pm}), \tag{12}$$

where $\lambda_i \leq -b$, $b > 0$ are the negative eigenvalues; $0 \leq \mu_i \leq a$, $a > 0$ are the positive eigenvalues and the ratio of positive to negative eigenvalues is a constant $p \in (0, 1/2)$. We can bound the r.h.s. of (11) with $(1 + 2bt)^{-\frac{(1-p)m}{2}}(1 - 2at)^{-\frac{pm}{2}}$. Elementary calculus shows that the minimum of this function over $t \in (0, 1/2a)$ gives:

$$P_r(v^t M v \geq 0) \leq \left( \frac{a^{1-p}b^p}{\frac{a+b}{2}} \frac{1}{2}(1-p)^{p-1}p^{-p} \right)^{\frac{m}{2}}, \tag{13}$$

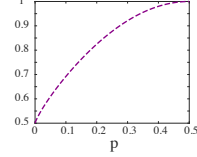

where $v$ is uniformly distributed on the unit sphere in $\mathbb{R}^m$. The function $\frac{1}{2}(1-p)^{p-1}p^{-p}$ is shown in the inset and for $p < 1/2$ it is strictly smaller than 1. The term $\frac{a^{1-p}b^p}{(a+b)/2}$ is the ratio of the weighted geometric mean and the arithmetic mean. Using the weighted arithmetic-geometric inequality it can be shown that these terms is at-most 1 if $a \leq b$. To summarize, if $a \leq b$ and $p < 1/2$ the probability to find a convex (positive) direction in $M$ is exponentially decreasing in $m$, the dimension of the matrix. One simple example is taking $a = b = 1$, $p = 0.49$ which shows that considering the matrices

$$U \big( \overbrace{-1, -1, \ldots, -1}^{0.51m}, \overbrace{1, 1, \ldots, 1}^{0.49m} \big) U^T$$

it will be extremely hard to get in random a convex direction in dimension $m \approx 300^2$, *i.e.*, the probability will be $\approx 4 \cdot 10^{-5}$ (this is a low dimension for a matching problem where $m = (n-1)^2$).

Another consequence that comes out of this theorem (in fact, its proof) is that the probability of finding a linear subspace $D \in G_r(d, m)$ for which the matrix $M$ is positive semidefinite is bounded by the probability of finding a one-dimensional subspace $D_1 \in G_r(1, m)$ to the power of $d$. Therefore the $d$ exponent in Definition 3 makes sense. Namely, to show a symmetric matrix $M$ is $\epsilon$-negative definite it is enough to check one-dimensional linear subspaces. An important implication of this fact and one of the motivations for Definition 3 is that finding local minima at high dimensional faces of the polytope $\mathrm{hull}(\mathcal{F})$ is much less likely than at low dimensional faces.

Next, we would like to prove Theorem 3 that shows that for natural probability space of Hessians $\{M\}$ the local minima of (10) are with high probability in $\mathcal{F}$, *e.g.*, permutations in case that $F = \Pi_n$. We therefore need to devise a natural probability space of Hessians. We opt to consider Hessians of the form discussed above, namely

$$\Omega_m = \left\{ U\Lambda_{m,p}U^T \mid U \in O(m) \right\}, \tag{14}$$

where $\Lambda_{m,p}$ is defined in (12). The probability measure over $\Omega_m$ is defined using the Haar probability measure on $O(m)$, that is for a subset $\mathcal{A} \subset \Omega_m$ we define $Pr(\mathcal{A}) = Pr(\{U \in O(m) \mid U\Lambda_{m,p}U^T \in \mathcal{A}\})$, where the probability measure on the r.h.s. is the probability Haar measure on $O(m)$. Note that (14) is plausible since the input graphs $G_A, G_B$ are usually provided with an arbitrary ordering of the vertices. Writing the quadratic energy $E$ resulted from a different ordering $P, Q \in \Pi_n$ of the vertices of $G_A, G_B$ (resp.) yields the Hessian $H' = (Q \otimes P)(B \otimes A)(Q \otimes P)^T$, where $Q \otimes P \in \Pi_m \subset O(m)$. This motivates defining a Hessian probability space that is invariant to $O(m)$. We prove:

**Theorem 3.** *If the number of extreme points of the polytope $\mathrm{hull}(F)$ is bounded by $\exp(m^{1-\epsilon})$, for some fixed arbitrary $\epsilon > 0$, and the Hessian of $E$ is drawn from the probability space $(\Omega_m, P_r)$, the chance that a local minimum of $\min_{X \in \mathrm{hull}(\mathcal{F})} E(X)$ is outside $\mathcal{F}$ is extremely small, bounded by $exp(-c_1 m)$, for some constant $c_1 > 0$.*

*Proof.* Denote all the edges (*i.e.*, one-dimensional faces) of the polytope $\mathcal{P} = \mathrm{hull}(\mathcal{F})$ by indices $\alpha$. Even if every two extreme points of $\mathcal{P}$ are connected by an edge there could be at most $\exp(2m^{1-\epsilon})$ edges. A local minimum $X_* \in \mathcal{P}$ to (10) that is not in $\mathcal{F}$ necessarily lies in the (relative) interior of some face $f$ of $\mathcal{P}$ of dimension at-least one. The restriction of the Hessian $M$ of $E(X)$ to $\mathrm{lin}(f)$ is therefore necessarily positive semidefinite. This implies there is a direction $v_\alpha \in \mathbb{R}^m$, parallel to an edge $\alpha$ of $\mathcal{P}$ so that $v_\alpha^T M v_\alpha \geq 0$.

Let us denote by $X_\alpha$ the indicator random variable that equals one if $v_\alpha^T M v_\alpha \geq 0$ and zero otherwise. If $X_\alpha = 1$ we say that the edge $\alpha$ is a *critical edge* for $M$. Let us denote $X = \sum_\alpha X_\alpha$ the random variable counting critical edges. The expected number of critical edges is $\mathbb{E}(X) = \sum_\alpha P_r(v_\alpha^T M v_\alpha \geq 0)$. We use Theorem 2, in particular (13), to bound the summands.

Since $P_r(v_\alpha^T M v_\alpha \geq 0) = P_r(v_\alpha^T U \Lambda_{m,p} U^T v_\alpha \geq 0)$ and $U^T v_\alpha$ is distributed uniformly on the unit sphere in $\mathbb{R}^m$, we can use (13) to infer that $P_r(v_\alpha^T M v_\alpha \geq 0) \leq \eta^{m/2}$ for some $\eta \in [0, 1)$ and therefore $\mathbb{E}(X) \leq \exp(m \log \eta / 2) \sum_\alpha 1$ (note that $\log \eta < 0$). Incorporating the bound on edge number in $\mathcal{P}$ discussed above we get $\mathbb{E}(X) \leq \exp(\frac{\log \eta}{2} m + 2m^{1-\epsilon}) \leq \exp(-c_1 m)$ for some constant $c_1 > 0$. Lastly, as explained above, the event of a local minimum not in $\mathcal{F}$ is contained in $X \geq 1$ and by Markov's inequality we finally get $P_r(X \geq 1) \leq \mathbb{E}(X) \leq \exp(-c_1 m)$. $\qquad\square$

Let us use this theorem to show that the local optimal solutions to Problem (10) with permutations as matchings, $\mathcal{F} = \Pi_n$, are with high probability permutations:

**Theorem 4.** *Let $E$ be a quadratic energy with Hessian drawn from the probability space $(\Omega_m, P_r)$. The chance that a local minimum of $\min_{X \in \mathrm{DS}} E(X)$ is outside $\Pi_n$ is extremely small, bounded by $exp(-c_1 n^2)$, for some constant $c_1 > 0$.*

*Proof.* In this case the polytope $\mathrm{DS} = \mathrm{hull}(\Pi_n)$ is in the $(n-1)^2$ dimensional linear subspace $\mathrm{lin}(\mathrm{DS})$ of $\mathbb{R}^{n \times n}$. It therefore makes sense to consider the Hessians' probability space restricted to $\mathrm{lin}(\mathrm{DS})$, that is considering $M|_{\mathrm{lin}(\mathrm{DS})}$ and the orthogonal subgroup acting on it, $O((n-1)^2)$. In this case $m = (n-1)^2$. The number of vertices of DS is the number of permutations which by Stirling's bound we have $n! \leq \exp(1 - n + \log n(n + 1/2)) \leq \exp((n-1)^{1.1})$. Hence the number of edges is bounded by $\exp(2(n-1)^{1.1})$, as required. $\qquad\square$

Lastly, Theorems 3 and 4, can be generalized by considering $d$-dimensional faces of the polytope:

**Theorem 5.** *If the number of extreme points of the polytope $\mathrm{hull}(F)$ is bounded by $\exp(m^{1-\epsilon})$, for some fixed arbitrary $\epsilon > 0$, and the Hessian of $E$ is drawn from the probability space $(\Omega_m, P_r)$, the chance that a local minimum of $\min_{X \in \mathrm{hull}(\mathcal{F})} E(X)$ is in the relative interior of a $d$-dimensional face of $\mathrm{hull}(\mathcal{F})$ is extremely small, bounded by $exp(-c_1 d m)$, for some constant $c_1 > 0$.*

This theorem is proved similarly to Theorem 3 by considering indicator variables $X_\alpha$ for positive semidefinite $M|_{\mathrm{lin}(\alpha)}$ where $\alpha$ stands for a $d$-dimensional face in $\mathrm{hull}(\mathcal{F})$. This generalized theorem has a practical implication: local minima are likely to be found on lower dimensional faces.

## 4   Graph matching with one sided permutations

In this section we examine an interesting and popular graph matching (1) instance, where the matchings are the one-sided permutations, namely $\mathcal{F} = \left\{ X \in \{0,1\}^{n \times n_0} \mid X\mathbf{1} = \mathbf{1} \right\}$. That is $\mathcal{F}$ are well-defined maps from graph $G_A$ with $n$ vertices to $G_B$ with $n_0$ vertices. This modeling is used in the template and partial matching cases. Unfortunately, in this case, standard graph matching energies $E(X)$ are not probably conditionally concave over $\mathrm{lin}(\mathcal{F})$. Note that $\mathrm{lin}(\mathrm{DS}) \subsetneq \mathrm{lin}(\mathcal{F})$.

We devise a variation of the Frank-Wolfe algorithm using a *concave search* procedure. That is, in each iteration, instead of standard line search we subtract a convex energy from $E(X)$ that is constant on $\mathcal{F}$ until we find a descent step. This subtraction is a relaxation of the original problem (1) in the sense it does not alter (up to a global constant) the energy values at $\mathcal{F}$.

The algorithm is summarized in Algorithm 2 and is guaranteed to output a feasible solution in $\mathcal{F}$. The linear program in each iteration over $\mathrm{hull}(\mathcal{F})$ has a simple closed form solution. Also, note that in the inner loop only $n$ different $\lambda$ values should be checked. Details can be found in the supplementary materials.

---

**input :** $X_0 \in \mathrm{hull}(\mathcal{F})$

**while** *not converged* **do**
    **while** *energy not reduced* **do**
        *add concave energy $M_{curr} = M - \lambda\Lambda$;*
        *compute step: $X_1 = \min_{X \in \mathrm{hull}(\mathcal{F})} [X_0]^T M_{curr} [X]$;*
        increase $\lambda$;
    **end**
    *Update current solution $X_0 = X_1$ and set $\lambda = 0$;*
**end**

**Algorithm 2:** Frank-Wolfe with a concave search.

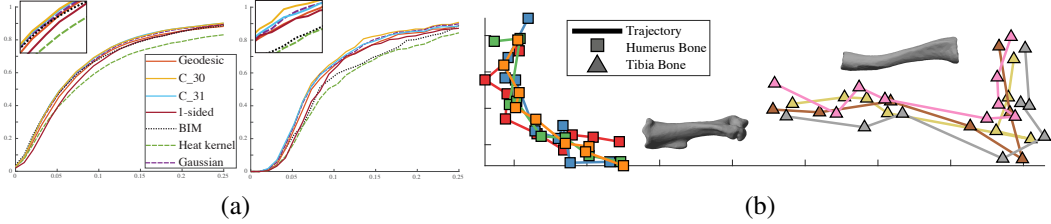

(a)                                                     (b)

Figure 1: (a) SHREC07 benchmark: Cumulative distribution functions of all errors (left) and mean error per shape (right). (b) Anatomical dataset embedding in the plane. Squares and triangles represent different bone types, lines represent temporal trajectories.

## 5   Experiments

**Bound evaluation:** Table 1 evaluates the probability bound (11) for Hessians $M \in \mathbb{R}^{100^2 \times 100^2}$ of $E_2(X)$ using affinities $A, B$ defined by functions of geodesic distances on surfaces. Functions that are conditionally negative definite or semi-definite in the Euclidean case: geodesic distances $d(x, y)$, its square $d(x, y)^2$, and multi-quadratic functions $(1 + d(x, y)^2)^{\frac{1}{10}}$. Functions that are positive definite in the Euclidean case: $c_{30}(\|x\|_2) = (1 - \|x\|_2)_+, c_{31}(\|x\|_2) = (1 - \|x\|_2)_+^4 (4\|x\|_2 + 1)$ and $\exp(-\tau^2\|x\|_2^2)$ (note that the last function was used in Vestner et al. (2017)). We also provide the *empirical* chance of sampling a convex direction. The results in the table are the mean over all the shape pairs (218) in the SHREC07 (Giorgi et al., 2007) shape matching benchmark with $n = 100$. The empirical test was conducted using $10^6$ random directions sampled from an i.i.d. Gaussian distribution. Note that 0 in the table means numerical zero (below machine precision).

Table 1: Evaluation of probable conditional concavity for different functions of geodesics on $\mathrm{lin}(\mathrm{DS})$.

|  | Distance | Distance Squared | MultiQuadratic | $c_{30}$ | $c_{31}$ | Gaussian |
|---|---|---|---|---|---|---|
| Bound mean | 0 | 0.024 | $7 \cdot 10^{-4}$ | 0 | 0 | 0 |
| Bound std | 0 | 0.021 | $1.7 \cdot 10^{-3}$ | 0 | 0 | 0 |
| Empirical mean | 0 | 0.003 | $7 \cdot 10^{-5}$ | 0 | 0 | 0 |
| Empirical std | 0 | 0.003 | $1.8 \cdot 10^{-4}$ | 0 | 0 | 0 |

**Initialization:** Motivated by Fischler and Bolles (1987); Kim et al. (2011) and due to the fast running time of the algorithms (*e.g.*, $150msec$ for $n = 200$ with Algorithm 1, and $16sec$ with Algorithm 2, both on a single CPU) we sampled multiple initializations based on randomized $l$-pairs of vertices of graphs $G_A, G_B$ and choose the result corresponding to the best energy. In Algorithm 1 we used the Auction algorithm (Bernard et al., 2016), as in Vestner et al. (2017).

Table 2: Comparison to "convex to concave" methods. The table shows the average and the std of the energy differences. Positive averages indicate our algorithm achieves lower energy on average.

|  | ModelNet10 | | | SHREC07 | | |
|---|---|---|---|---|---|---|
| # points | 30 | 60 | 90 | 30 | 60 | 90 |
| DSPP | 5.0± 5.3 | 9.8± 10.8 | 14.468± 19.8 | 1.3± 2.3 | 9.5± 9.5 | 26.2± 24.3 |
| PATH | 101.4±53.9 | 512.3±198.4 | 1251.9±426.4 | 69.263±55.9 | 307.7±230.6 | 721.0±549.7 |
| RANDOM | 197.9±35.2 | 865.3±122.1 | 1986.1±273.0 | 120.2±83.6 | 532.7±357.8 | 1230.7±817.6 |

**Comparison with convex-to-concave methods:** Table 2 compares our method to Zaslavskiy et al. (2009); Dym et al. (2017) (PATH, DSPP accordingly). As mentioned in the introduction, these methods solve convex relaxations and then project its minimizer while deforming the energy towards concavity. Our method compares favorably in the task of matching point-clouds from the ModelNet10 dataset (Wu et al., 2015) with Euclidean distances as affinities, and the SHREC07 dataset (Giorgi et al., 2007) with geodesic distances. We used $\mathcal{F} = \Pi_n$, and energy (3). The table shows average and standard deviation of energy differences of the listed algorithms and ours; the average is taken over 50 random pairs of shapes. Note that positive averages mean our algorithm achieves lower energy on average; the difference to random energy values is given for scale.

**Automatic shape matching:** We use our Algorithm 1 for automatic shape matching (*i.e.*, with no user input or input shape features) on a the SHREC07 (Giorgi et al., 2007) dataset according to the protocol of Kim et al. (2011). This benchmark consists of matching 218 pairs of (often extremely) non-isometric shapes in 11 different classes such as humans, animals, planes, ants etc. On each shape, we sampled $k = 8$ points using farthest point sampling and randomized $s = 2000$ initializations of subsets of $l = 3$ points. In this stage, we use $n = 300$ points. We then up-sampled to $n = 1500$ using the exact algorithm with initialization using our $n = 300$ best result. The process takes about $16min$ per pair running on a single CPU. Figure 1 (a) shows the cumulative distribution function of the geodesic matching errors (left - all errors, right - mean error per pair) of Algorithm 1 with geodesic distances and their functions $c_{30}, c_{31}$. We used (3) and $\mathcal{F} = \Pi$. We also show the result of Algorithm 2 with geodesic distances, see details in the supplementary materials. We compare with Blended Intrinsic Maps (BIM) (Kim et al., 2011) and the energies suggested by Boyarski et al. (2017) (heat kernel) and Vestner et al. (2017) (Gaussian of geodesics). For the latter two, we used the same procedure as described above and just replaced the energies with the ones suggested in these works. Note that the Gaussian of geodesics energy of Vestner et al. (2017) falls into the probably concave framework.

**Anatomical shape space analysis:** We match a dataset of 67 mice bone surfaces acquired using micro-CT. The dataset consists of eight time series. Each time series captures the development of one type of bone over time. We use Algorithm 1 to match all pairs in the dataset using Euclidean distance affinity matrices $A, B$, energy (3), and $\mathcal{F} = \Pi_n$. After optimization, we calculated a $67 \times 67$ dissimilarity matrix. Dissimilarities are equivalent to our energy over the permutations (up to additive constant) and defined by $\sum_{ijkl} X_{ij} X_{kl} (d_{ik} - d_{jl})^2$. A color-coded matching example can be seen in the inset. In Figure 1 (b) we used Multi-Dimensional Scaling (MDS) (Kruskal and Wish, 1978) to assign a $2D$ coordinate to each surface using the dissimilarity matrix. Each bone is shown as a trajectory. Note how the embedding separated the two types of bones and all bones of the same type are mapped to similar time trajectories. This kind of visualization can help biologists analyze their 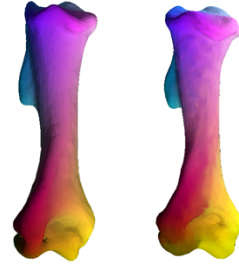 data and possibly find interesting time periods in which bone growth is changing. Lastly, note that the Tibia bones (on the right) exhibit an interesting change in the midst of its growth. This particular time was also predicted by other means by the biologists.

# 6  Conclusion

In this work, we analyze and generalize the idea of concave relaxations for graph matching problems. We concentrate on *conditionally concave* and *probably conditionally concave* energies and demonstrate that they provide useful relaxations in practice. We prove that all local minima of such relaxations are with high probability in the original feasible set; this allows removing the standard post-process projection step in relaxation-based algorithms. Another conclusion is that the set of optimal solutions of such relaxations coincides with the set of optimal solutions of the original graph matching problem.

There are popular edge affinity matrices, such as $\{0, 1\}$ adjacency matrices, that in general do not lead to conditionally concave relaxations. This raises the general question of characterizing more general classes of affinity matrices that furnish (probably) conditionally-concave relaxations. Another interesting future work could try to obtain information on the quality of local minima for more specific classes of graphs.

# 7  Acknowledgments

The authors would like to thank Boaz Nadler, Omri Sarig, Vova Kim and Uri Bader for their helpful remarks and suggestions. This research was supported in part by the European Research Council (ERC Consolidator Grant, "LiftMatch" 771136) and the Israel Science Foundation (Grant No. 1830/17). The authors would also like to thank Tomer Stern and Eli Zelzer for the bone scans.

## Footnotes

[1]In a nutshell, positive (negative) definite functions are functions that when applied to differences of vectors produce positive (negative) definite matrices when restricted to certain linear subspaces; this notion will be formally introduced and defined in Section 2.

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
