[Supplementary Material]

# (Probably) Concave Graph Matching
# Supplementary material

**Haggai Maron**
Weizmann Institute of Science
Rehovot, Israel
`haggai.maron@weizmann.ac.il`

**Yaron Lipman**
Weizmann Institute of Science
Rehovot, Israel
`yaron.lipman@weizmann.ac.il`

## 1   Frank-Wolfe with concave search

An orthogonal basis to $\mathrm{lin}(\mathcal{F})$ is computed similarly to Lemma 1 in the paper:

**Lemma 1** (orthonormal basis for one-sided permutations). *If the columns of $F \in \mathbb{R}^{n_0 \times (n_0-1)}$ form an orthonormal basis for $\mathbf{1}^\perp$ in $\mathbb{R}^{n_0}$ then the columns of $F \otimes I_n$ are an orthonormal basis for $\mathrm{lin}(\mathcal{F})$.*

The energy $E_2(X)$ in this case does not model the matching problem well since it gives rise to trivial solutions. Instead, we chose to optimize a similar energy (Solomon et al., 2016): $E(X) = \sum_{ijkl} X_{ij} X_{kl} (A_{ik} - B_{jl})^2$. This energy can also be written in matrix form: $[X]^T M [X]$ where $M = -2B \otimes A + 11^T \otimes A.^2 + B.^2 \otimes 11^T$ (where $C.^2$ implies entry-wise operation) and after restricting it to $\mathrm{lin}(\mathcal{F})$ its Hessian is of the form $-2FBF \otimes A + FB.^2 F \otimes 11^T$. Assuming $A, B$ are Euclidean distance matrices, the right summand is negative semidefinite, but the left summand is not. This is because that $A$ is not conjugated by $F$: it has a large positive eigenvalue as a result of the Perron-Frobenius Theorem.

The linear program solved in each iteration of the algorithm takes a surprisingly simple form: it amounts to solving $\min_{X \in \mathrm{hull}(\mathcal{F})} \mathrm{tr}(\nabla E(X_0)^T X)$ which can be solved simply by assigning the value 1 to the index of the minimal value in each row of $\nabla E(X_0)$. This procedure always outputs solutions in $\mathcal{F}$.

The convex energies we subtract from the objective during the concave search should be constant on $\mathcal{F}$ so a reduction in the subtracted energy is the same as in the original energy $E(X)$. We use the quadratic form defined by $\lambda * \Lambda$ where $\Lambda$ is a $nn_0 \times nn_0$ diagonal matrix defined by $D_{ijij} = \max_j \{\sum_{kl} |M_{ijkl}|\}$. $D$ is a positive definite matrix and for $\lambda = 1$, $W - D$ is guaranteed to be negative semidefinite. The values of $\lambda$ need not be discretized since there are only $n$ different critical values - the ones that change the minimum calculation mentioned in the previous paragraph.

## References

Solomon, J., Peyré, G., Kim, V. G., and Sra, S. (2016). Entropic metric alignment for correspondence problems. *ACM Transactions on Graphics (TOG)*, 35(4):72.