[Reviews · NeurIPS 2018]

Reviewer 1



Update: I have considered the author response. Some of my previous questions have been answered. I have updated my review and "overall score". ========================================================== This paper focuses on graph matching problems with quadratic objective functions. These objectives are usually equivalent to a quadratic assignment problem (QAP) [1] on the permutation matrix space. The authors focused on the Frank-Wolfe algorithm (Algorithm 1) on the objectives with the relaxation from the permutation matrix space to the doubly stochastic matrix space DS. To analyze the performances of this algorithm, the authors defined conditionally concave (or negative definite) and used these definitions to do some theoretical analysis on the local optima convergence of this algorithm. Then the authors extended this definition to a probabilistic setting and did some analysis for a more general form of graph matching problems. At the end, the authors provided another algorithm (Algorithm 2) for the one sided permutation problems where the probably conditionally concavity does not hold. Graph matching problems are very popular now in the machine learning community. In my opinion, the authors proposed some very interesting and novel definitions, and these definitions may be very useful for doing relaxations for graph matching problems. However, it will be better if more analysis on the optimality of the convergence local optima can be shown instead of just showing some sufficient conditions for the algorithm to guarantee (or with high probability) to converge to extreme point local optima. Strengths of this paper: 1. Proposed the definition for (probably) conditionally concave and (probably) conditionally negative definite. These definitions can clearly capture the possibility for the Frank-Wolfe algorithm to locally optimize the objective function. 2. The theoretical analysis on these new concavity definitions provide us important insight on the convergence performances of Algorithm 1, under certain circumstances. Weakness: 1. As I just mentioned, the paper only analyzed, under which cases will the Algorithm 1 converges to permutations as local minima. However, it will be better if the quality of this kind of local minima could be analyzed (e.g. the approximation ratio of these local minima, under certain assumptions). 2. This paper is not very easy to follow. First, many definitions are used before they are defined. For example, on line 59 in Theorem 1, the authors used the definition of "conditionally positive definite function of order 1", which is defined on line 126 in Definition 2. Also, the author used the definition "\epsilon-negative definite" on line 162, which is defined on line 177 in Definition 3. It will be better if the authors could define those important concepts before using them. Second, the introduction is a little bit too long (more than 2.5 pages) and many parts of that are repeated in Section 2 and 3. It might be better to restructure the first 3 sections for a little bit. Third, it will be good if more captions could be added to the figures in the experiment section so that the readers could understand the results more easily. 3. For the description of Theorem 3, from the proof it seems that we need to use Equation 12 as a condition. It will be better if this information is included in the theorem description to avoid confusions (though I know this is mentioned right before the theorem, it is still better to have it in the theorem description). Also, for the constant c_1, it is better to give it an explicit formula in the theorem. Reference: [1] Burkard, R. E., Cela, E., Pardalos, P. M., and Pitsoulis, L. S. (1998). The quadratic assignment problem. In Handbook of combinatorial optimization, pages 1713–1809. Springer.

Reviewer 2



In this paper, the authors analize the graph matching problem, with special focus on concave relaxations. The paper is very well organized. I really liked the introduction, which is like an extended abstract. It is well written in general, it's didactical and it's easy to follow. Good work. The main contributions are in the study of conditional concave energies (when restriced to the linear subspace resulting from the convex hull of the matching set), where they prove that certain family of graphs fall in this categoty, and then the stochastic version, where they study these properties in probability. The math seems to be ok, and it's somehow elegant. I would have liked some experimets with "traditional" graphs, i.e., undirected unweighted graphs (with binary symmetric adjacency matrix), since the fact that the matrices are binary, may break some hipothesis (I'm thinking in the Hessian drawn from some probability space), and it would be interesting to see the results in practice. I really liked the paper in general. Regarding the bibliography: The ds++ paper is published in ACM Transactions on Graphics (the paper refers to the arxiv version). Also, there are three other papers that may have some common points with this work. The classical Umeyama work: An eigendecomposition approach to weighted graph matching problems. PAMI 1988 And two works regarding the convex hull relaxation: - A linear programming approach for the weighted graph matching problem, HA Almohamad, SO Duffuaa, PAMI, 1993. - On spectral properties for graph matching and graph isomorphism problems. Fiori and Sapiro, Information and Inference, 2015.

Reviewer 3



This paper deals with the in general NP-hard graph matching problem, which has been intensively studied over years, whereby many ad-hoc empirical methods are devised. While this paper gives a fresh and important anlysis to some special cases (but often common in practice) and lead to effective graph matching algorithms. This paper is well written and the information is very intensive, whereby closely related works are well covered. Pros: 1) prove there is a large class of affinity matrices resulting in conditionally concave 2) generalize the notion of conditionally concave energies to probably conditionally concave energies, the authors further show the fact that probably conditionally concave energies are very common in many affinity matrix. This suggests the usefulness of the proposed method derived from these findings. 3) for the case of not probably conditionally concave, the authors devise a variation of the Frank-Wolfe algorithm using a concave search procedure wherenby concave search is used to replace the standard line search 4) the experimental results on multiple datasets are very impressive, and the improvement is notable and consistent Minor comments: Reference [Vestner et al., 2017] does not contain the venue information, it is a CVPR 2017 paper. In the final version, it would be nice to add a conclusion section in the end whereby possible future work can also be discussed. I suggest Table 2 caption shall contain more information about the performance measurement i.e. average and standard deviation of energy differences, not only stated in the main text. Also the algorithm name in table 2 shall be explicitly mentioned in the main text.